# A longitudinal study of the association between attending cultural events and coronary heart disease

Sven-Erik Johansson[1], Filip Jansåker [1,2✉], Kristina Sundquist[1,3,4] & Lars Olov Bygren[5,6]

## Abstract

**Background** The experiences of art and music are an essential part of human life and this study aimed to examine the longitudinal association between cultural participation and coronary heart disease.

**Methods** This was a longitudinal study on a randomly selected representative adult cohort ($n = 3296$) of the Swedish population. The study period was over 36 years (1982–2017) with three separate eight-year interval measurements of cultural exposure (for example, visiting theatres and museums) starting in 1982/83. The outcome was coronary heart disease during the study period. Marginal structural Cox models with inverse probability weighting were used to account for time-varying weights of the exposure and potential confounders during the follow-up. The associations were also examined through a time-varying Cox proportional hazard regression model.

**Results** Cultural participation shows a graded association, the higher the exposure the lower the risk of coronary heart disease; the hazard ratio was 0.66 (95% confidence interval, 0.50 to 0.86) for coronary heart disease in participants with the highest level of cultural exposure compared with the lowest level.

**Conclusion** Although causality cannot be determined due to the remaining risk of residual confounding and bias, the use of marginal structural Cox models with inverse probability weighting strengthens the evidence for a potentially causal association with cardiovascular health, which warrants further studies.

## Plain language summary

This study examined whether people taking part in cultural activities, such as going to museums or theatres, were less likely to get coronary heart disease. The study included 3296 adults in Sweden over a period of 36 years. Information on cultural participation was collected by questionnaires on three occasions, eight-years apart. National healthcare data was used to identify cases of coronary heart disease. The main finding was that people who took part in more cultural activities were less likely to have coronary heart disease. This study suggests that taking part in cultural activities may be an important way to keep your heart healthy.

[1] Center for Primary Health Care Research, Department of Clinical Sciences Malmö, Lund University, Malmö, Sweden. [2] Department of Clinical Microbiology, Rigshospitalet, Copenhagen University Hospital, Copenhagen, Denmark. [3] Department of Family Medicine and Community Health, Department of Population Health Science and Policy, Icahn School of Medicine at Mount Sinai, New York, USA. [4] Center for Community-based Healthcare Research and Education (CoHRE), Department of Functional Pathology, School of Medicine, Shimane University, Shimane, Japan. [5] Community Medicine and Rehabilitation, Umeå University, Umeå, Sweden. [6] Department of Biosciences and Nutrition, Karolinska Institutet, Huddinge, Sweden. ✉email: filip.jansaker@med.lu.se

The experiences of art and music are an essential part of human life, and their association with health and longevity has been a subject of study in many sciences. In philosophy, in his most extensive and basic contribution[1], the American philosopher, John Dewey, hinted the mechanism behind the potential effect of art on human wellbeing. The experience of art is likely not only a mere event (in German terms: erfarung) but an overwhelming feeling (erlebung)[1] and having such experiences at a young age might affect subsequent generations[2]. In psychology, the evidence on leisure's strengthening effect on health has been accumulating[3] and animal trials have found an enriched environment to be associated with learning and health in mice[2,4]. In medicine, attendance at fine arts events was found to be inversely associated with all-cause mortality by our group over two decades ago[5], which has been replicated in several studies over the intervening years[6–8]. However, the potentially causal relationship between cultural activities and the most common causes of mortality[9,10] has not yet been established. This is likely because risk factors often vary over time and may have acted as residual confounding factors in the previous studies on cultural exposure and health due to the limitations of more traditional epidemiological designs[5–7]. Therefore, comprehensive studies exploring potential causal associations between cultural activities and health are needed. Such studies could provide a more contextual understanding of the epidemiological pathways between cultural exposure and health. This can help policymakers allocate resources in society to evidence-based health-promoting activities.

Marginal structural modelling Cox (MSM–Cox) with inverse probability weights (IPW) is an established method to estimate causal treatment effects in observational studies and provides a more robust assessment than conventional Cox models in the presence of time-varying variables, given that all assumptions are satisfied[11–15]. For example, MSM-Cox has been used to estimate the causal average treatment effect from observational data similar to that of randomised controlled trials[12] in order to minimise, albeit not eliminate, the risk of residual confounding and certain forms of selection bias[11]. By applying this method, we aimed to assess the potential causal association between attending cultural events and coronary heart disease (CHD)—the leading cause of mortality worldwide[9,10]. The study included 3296 adults in Sweden, with three measurements (8-years apart, starting in 1982/1983) and follow-up until the end of 2017. We found an inverse association between cultural participation and CHD, which may help to elucidate potential mechanisms that may be found in the complex backgrounds of leisure time including, on one hand, attending cultural events and, on the other, matching the complexity of CHD risk factors.

## Methods

**Study design and setting.** An observational study of longitudinal design with three separate measurements at eight-year intervals. The participants were followed from the year 1981 or 1982 (baseline) until the year 2017 (end of the study period). The STROBE statement checklist for cohort studies was considered when conducting the study and writing the manuscript[16]. The study was conducted at the Department of Clinical Sciences Malmö, Lund University, Sweden.

**Ascertainment of the study population.** The study population consisted of a longitudinal subgroup of participants (from a random sample of the Swedish population) collected from the national survey on living conditions in Sweden (In Swedish, Undersökningarna av Levnadsförhållanden, ULF)[17]. To minimise residual confounding and capitalise as much as possible on the use of time-varying variables, only participants with three measurements and without a diagnosis of CHD occurring before the start of the study period were included in the analysis. A total of 3311 participants had completed all three measurements in ULF, of which 15 were excluded due to a CHD diagnosis prior to baseline, leaving a total of 3296 participants for inclusion in the analysis. Data were split into 1-year intervals and originated from the following years: 1982/83, 1990/91, and 1998/99; the follow-up ended on 31 December 2017.

**Ascertainment of cultural exposure.** Cultural exposure was defined using self-reported data on attending cultural events at three measurement points, collected in ULF[17]. The outcome was defined as the rate of visiting (from never to every week or more often) art- and other museums, cinemas, concerts, sermons and theatres. To define the level of cultural exposure at the three measurement periods we calculated a cultural attendance index based on the factor scores from a Principal factor analysis (polychoric correlation coefficient for ordinal data). The results from this index were categorised separately for each of the three measurements into lowest (25%, quartile 1), middle (50%, quartiles 2–3) and highest (25%, quartile 4) level of cultural exposure. These categories were chosen because they were judged to be useful and illustrative in the interpretation of the results. The lowest level of cultural exposure was used as the reference group (Hazard Ratio, HR = 1), i.e., in the analyses, the participants with the highest and middle level of cultural exposure were compared with the lowest level. The analysis took into consideration any potential change in exposure to specific cultural activities to create a unique cultural attendance index at each of the three measurement periods.

**Ascertainment of the outcome.** The outcome in this study was incidence of ischaemic/coronary heart disease (in this study: CHD) defined according to the 9th and 10th international classification of diseases (ICD), i.e., I20–I25 (ICD-10) from 1997 or 410–414 (ICD-9) prior to 1997.

**Ascertainment of the potential confounders.** All potential confounding variables used in this study were collected from self-reported data in ULF[17] at three separate measurement points, except for region of residence which was collected from nationwide population register data (see below). Age was defined in years (mean). Educational level was categorised as low (≤9 years), middle (10–12 years) and high (≥13 years) level of schooling. Marital status was defined as having responded to be married/cohabiting or single living. Social isolation was grouped as socially isolated (responding "has no close friend") or not socially isolated (responding "has a close friend"). Physical exercise was defined as inactive/occasionally active or actively exercising at least once a week indoors or outdoors. Region of residence was categorised into large cities, middle-large cities and all other regions. Tobacco smoking was defined as daily smoking or not (the latter including past use). Sex was collected at baseline as being a woman or a man (and did not change during the three measurements). The potential interaction between sex and age was also measured, found to be significant and taken into account in the analyses[18].

**Data sources.** The national survey on living conditions, i.e., The Statistics on Income and Living Conditions (SILC or ULF in Swedish, which is the abbreviation used in this paper), is an annual survey performed by the governmental authority on national statistics, Statistics Sweden (Swedish: Statistikmyndigheten, SCB)[17]. This data source contains information on living conditions collected from randomly selected individuals of the

Swedish population aged 16 and older. The survey is repeated for a random proportion of individuals from 1982/83 and onwards. The diagnoses of the outcome CHD were collected through almost nationwide (Swedish) primary healthcare data (1990–2017) and the National Patient Register[19] with almost complete outpatient- (specialist care) (2001–2018) and inpatient (hospital) data (1964–2018); similar to previous studies by our group[15]. The coverage of the primary healthcare data was collected from 20 out of 21 Swedish administrative regions. The National Patient Register is managed by the Swedish National Board of Health and Welfare (Swedish: Socialstyrelsen). Mortality and emigration data were collected from the Cause of Death Registers (Socialstyrelsen, 1961–2018) and the Total Population Register (SCB, 1968–2018), which were nearly complete for the entire national population. The latter register was also used to collect data on region of residence based on information provided to us by Statistics Sweden. All linkages between the various sources of data were done by using the unique 10-digit personal identification number assigned to each person for their lifetime upon birth or immigration to Sweden. This code was replaced by a pseudonymised version of this number to protect the integrity of all individuals.

**Statistics and reproducibility**. The distribution of the included variables by period and level of cultural exposure were presented as proportions and means. Incidence rates of CHD per 1000 person-years for the included variables were also estimated by Poisson regression, adjusted for sex and age. The study used both MSM-Cox models with IPW (based on time-varying weights) and Cox proportional hazard regression models (based on time-varying variables), which were used to estimate HRs for CHD in participants in relation to cultural exposure, adjusted for the selected confounders. All variables were allowed to be time-varying in the analysis expect for sex (which did not change in our sample during the follow-up period). A two-tailed *p* value of <0.05 was used to determine statistical significance.

The MSM-Cox (with time-varying weights, IPW) was used to take residual confounding due to time-varying exposure into account as it allowed cultural exposure and the covariates, except for sex, to vary over time. This method provides marginal HR estimations, which has been used for more robust causal interpretations in observation studies[11–15], albeit not entirely eliminating confounding and other sources of bias. The time-varying IPW was estimated using a logistic model that regresses covariates to a treatment group (exposure) variable. The analysis used marginal approach and weights to balance the confounders across exposure levels. Propensity scores were used on repeated measurements to calculate the IPW[13] and used to assess the inverse of the probabilities of cultural exposure at each time point during the follow-up. The MSM-Cox requires three essential assumptions to be fulfilled for a causal interpretation. Firstly, all possible confounders must be included. Secondly, a positivity assumption that every participant must have at least a non-zero probability to be exposed, *i.e.*, in this study cultural activities. This means that none of the predicted values that are used to compute the propensity scores and the IPWs ought to be 0 or 1[20]. Thirdly, the method should be able to include a correct specification of the IPW model in the analysis[14]. In the present study, these three necessary assumptions for a causal association were judged to be approximately satisfied in the following ways: Firstly, although in a theoretically ideal setting, all possible confounders should be considered. The confounding variables between cultural exposure and CHD has not been well-studied but the most important potential confounders were selected based on data from previous studies on cultural exposure and health[5,21]. Secondly, all

participants were assumed to have a non-zero theoretical probability to be exposed to cultural events. Thirdly, the IPW model was done in accordance with established methods. In addition, both death and emigration were taken into account when calculating time at risk. We also included the main effects of sex and age as well as the significant interaction between sex and age when we estimated the IPW. The weights (IPW) were smooth and low with satisfactory dispersion and the median and mean approximately equal (Min. = 0.33; 1st Qu. = 0.73; Median = 0.92; Mean = 1.00; 3rd Qu. = 1.06; and Max. = 14.42).

In the time-varying Cox proportional hazard regression model, the analysis was adjusted for all covariates used in the IPW. If a variable did not satisfy the proportional assumption in the Cox proportional hazard regression, it was included as a stratum in the model. This resulted in equal coefficients across the strata but with a baseline hazard distinct for each stratum. The analysis obtained no HRs for stratified variables. Schoenfeld residuals were used to test proportionality (calculated and reported only at failure times).

We used the statistical software R[20] for analyses in the MSM-Cox and STATA[22] for analyses in the Cox proportional hazard regression. The IPW R-package[23] was used to estimate IPW for measurements in the longitudinal data. The statistical code used to obtain the results can be found in Supplementary Data.

**Ethical statement**. This study was a non-intervention register study on already collected and encrypted secondary data. It was conducted according to the guidelines of the Declaration of Helsinki and approved by the Ethical Review Board in Lund and was exempted from informed consent requirements owing to its register-based design. Access to the used registries was obtained from Swedish authorities prior to the study commencing and all methods were used in accordance with national guidelines and regulations.

**Patient and public involvement statement**. The public and patients were not involved in this study regarding the general concept, idea and design.

**Reporting summary**. Further information on research design is available in the Nature Portfolio Reporting Summary linked to this article.

## Results

Table 1 shows the distribution and variation over time of the covariates based on the level of cultural exposure. Table 2 shows the incidence rates of CHD per 1000 person-years for cultural exposure and each covariate. The participants with the lowest level of cultural exposure had the highest incidence rate of CHD. Participants who had reported daily tobacco smoking and low education level also had notably high incidence rates, while participants with a high level of education and cultural exposure had low incidence rates.

Table 3 includes the estimated HRs with 95% confidence intervals (CI) for CHD associated with cultural exposure, which presents the Cox proportional hazard regression model (adjusted for all confounders) and the MSM-Cox model (with IPW estimated by a multinomial model containing all confounders). The MSM-Cox model shows a suggested gradient towards decreased incidence of CHD in the participants by increasing cultural exposure. For example, the participants with a high level of cultural exposure had an ~34% risk reduction for CHD compared to the participants with the lowest. The Cox proportional hazard regression model showed a significant, relative risk reduction in CHD of 26% only for a high level of cultural exposure.

**Table. 1 The distribution (percentages and mean) of the included confounders for the participants by time of measurement and level of cultural exposure.**

| Variable | Level | 1982/83 (n = 3296) | | | 1990/91 (n = 3296) | | | 1998/99 (n = 3296) | | |
|---|---|---|---|---|---|---|---|---|---|---|
| Cultural exposure | n | Low | Middle | High | Low | Middle | High | Low | Middle | High |
| Number | | 813 | 1659 | 824 | 812 | 1641 | 843 | 729 | 1729 | 838 |
| Sex | Men | 50.8 | 48.8 | 43.1 | 54.2 | 47.7 | 42.1 | 52.3 | 48.6 | 42.5 |
| | Women | 49.2 | 51.2 | 56.9 | 45.8 | 52.3 | 57.9 | 47.7 | 51.4 | 57.5 |
| Age | (years) | 42.5 | 39.2 | 38.3 | 49.7 | 47.3 | 46.9 | 62.6 | 54.1 | 53.5 |
| Region of residence | The largest cities | 23.3 | 28.8 | 44.3 | 23.2 | 28.9 | 42.5 | 22.1 | 29.4 | 39.5 |
| | Middle-large towns | 34.4 | 31.0 | 27.7 | 34.9 | 35.6 | 33.8 | 36.6 | 36.6 | 35.1 |
| | All others | 42.3 | 40.2 | 28.0 | 41.9 | 35.5 | 23.7 | 41.3 | 34.0 | 25.4 |
| Education level | Low (≤9 years) | 85.4 | 72.8 | 43.5 | 81.9 | 68.5 | 38.3 | 86.3 | 66.8 | 34.7 |
| | Middle (10–12 years) | 12.6 | 20.7 | 35.1 | 16.3 | 23.2 | 34.2 | 11.3 | 24.6 | 37.5 |
| | High (≥13 years) | 2.0 | 6.6 | 21.4 | 1.8 | 8.3 | 27.5 | 2.5 | 8.6 | 27.8 |
| Marital status | Married/cohabiting | 78.7 | 75.9 | 62.9 | 76.6 | 78.9 | 76.2 | 63.8 | 75.1 | 76.9 |
| | Single living | 21.3 | 24.1 | 37.1 | 23.4 | 21.1 | 23.8 | 36.2 | 24.9 | 23.1 |
| Tobacco smoking | Daily | 39.1 | 30.6 | 26.3 | 33.4 | 26.5 | 21.8 | 24.1 | 20.8 | 14.3 |
| | Not/Past use | 60.9 | 69.4 | 73.7 | 66.6 | 73.5 | 78.2 | 75.9 | 79.2 | 85.7 |
| Physical exercise | At least once a week | 22.0 | 42.0 | 59.7 | 21.3 | 34.4 | 49.0 | 16.6 | 34.0 | 49.2 |
| | Less than once a week | 78.0 | 58.0 | 40.3 | 78.7 | 65.6 | 51.0 | 83.4 | 66.0 | 50.8 |
| Socially isolated | Yes | 25.7 | 22.0 | 31.3 | 28.8 | 27.1 | 28.1 | 44.9 | 33.7 | 30.9 |
| | No | 74.3 | 78.0 | 58.7 | 71.2 | 72.9 | 71.9 | 54.1 | 66.3 | 69.1 |

Level of cultural exposure at each measurement period: *low* the lowest quartile of cultural exposure; *middle* the two middle quartiles of cultural exposure; and *high* the highest quartile of cultural exposure.

**Table 2 Sex- and age-adjusted incidence rates per 1000 person-years of coronary heart disease (n = 3296) and number of events (cases = 694) by the level of cultural exposure and the included confounders.**

| Variables | Level | Incidence rates[1] (events) | 95% confidence intervals | P values[2] |
|---|---|---|---|---|
| Attending cultural events[3] | Low level | 7.8 (211)[4] | 6.7–8.8 | Ref. |
| | Middle level | 7.1 (358) | 6.3–7.8 | 0.22 |
| | High level | 5.0 (125) | 4.2–6.0 | 0.000 |
| Region of residence | The largest cities | 6.3 (193) | 5.4–7.2 | Ref |
| | Middle-large towns | 6.9 (249) | 6.0–7.8 | 0.39 |
| | All others | 7.0 (252) | 6.1–7.9 | 0.28 |
| Education | Low (≤9 years) | 7.4 (523) | 6.7–8.0 | 0.001 |
| | Middle (10–12 years) | 5.6 (122) | 4.5–6.5 | 0.17 |
| | High (≥13 years) | 4.7 (049) | 3.4–6.0 | Ref. |
| Marital status | Married/cohabiting | 6.6 (497) | 6.0–7.2 | Ref. |
| | Single living | 7.2 (197) | 6.2–8.2 | 0.26 |
| Tobacco smoking | Daily | 7.6 (136) | 6.3–8.9 | 0.26 |
| | Not/Past use | 6.6 (558) | 6.0–7.1 | Ref. |
| Physical exercise | At least once a week | 6.1 (187) | 5.2–7.0 | 0.10 |
| | Less than once a week | 7.0 (507) | 6.4–7.7 | Ref. |
| Socially isolated | Yes | 6.8 (259) | 6.0–7.7 | Ref. |
| | No | 6.7 (435) | 6.1–7.3 | 0.58 |

[1]Estimated by Poisson regression adjusted for sex and age (categorised in 8-year bands), shown as incidence rates with 95% confidence intervals.
[2]P values within each level of the variables compared with a reference category in a sex-and age-adjusted Poisson model.
[3]Level of cultural exposure: *Low* the lowest quartile, *Middle* the two middle quartiles, and *High* the highest quartile. The level of cultural exposure was based on an index score from a Principal factor analysis (including the variables: visiting cinema, theatre, concert, art museum, other museums and sermons).
[4]Number of events.

## Discussion

This longitudinal study attempted to assess a possible causal association between attending cultural events and CHD. The main innovations behind the findings were that the methodological approach included the use of MSM-Cox with IPW, complete data from a random sample of the Swedish population with three separate repeated measurements and a very long follow-up period (36 years).

The findings suggested a gradient of decreased incidence rates of CHD among adults when cultural exposure increased. The estimates based on MSM-Cox with IPW were somewhat stronger than in the Cox proportional hazard regression model (which adjusted for the same variables included in IPW). This may be attributed to the use of weights in the MSM-Cox models with IPW that may further minimise, albeit not eliminate, the risk of residual confounding due to time-dependent covariates, which may act as both confounders and mediators[11–14,24]. Taken together with previous plausibility suggestions[1,3], coherence with animal data[2,4] and concordance with conventional observational studies[5–7], the present study used a unique, longitudinal data source. Leisure time attendance at cultural events and CHD are complex in their potentially causal association but may encompass environmental factors capable of influencing, for example, epigenetic mechanisms related to cardiovascular diseases and type 2 diabetes[25–28]; thus, potential mechanisms might lie in the environmental influences on genomics, i.e., epigenetics[27,29].

**Table. 3 The association between cultural exposure and coronary heart disease in a longitudinal study of three separate measurements with eight-year intervals (n = 3296).**

| Exposure of attending cultural events[1] | Cox proportional hazard regression[2] | MSM-Cox with IPW[3] |
|---|---|---|
| | Hazard ratio (95% confidence interval) | |
| Low level | Reference | Reference |
| Middle level | 0.95 (0.79–1.13) p value = 0.49 | 0.80 (0.66–0.98) p value = 0.029 |
| High level | 0.74 (0.58–0.95) p value = 0.009 | 0.66 (0.50–0.86) p value = 0.002 |

[1]Level of cultural exposure: *Low* the lowest quartile, *Middle* the two middle quartiles, *High* the highest quartile. The level of cultural exposure was based on an index score from a Principal factor analysis (including the variables: visiting cinema, theatre, concert, art museum, other museums and sermon)[2]. Adjusted for sex, age, the interaction of sex and age, main region, education, marital status, smoking, exercise and socially isolated[3]. Marginal Structural Modelling Cox (MSM-Cox) with inverse probability weights (IPW), based on the confounding variables above.

There are some limitations to this study. Firstly, although it included several of the potentially important confounders[5,21], including several important risk factors related to CHD[30,31], we cannot fully rule out the possibility of residual confounding. For example, we had no information on some of the most important risk factors for CHD (e.g., dyslipidaemia, obesity, family risk and diabetes mellitus)[15,30,31], since confounding variables between cultural exposure and CHD have only been sparsely studies[5,21]. Therefore, we cannot determine whether all assumptions for MSM-Cox were fulfilled for a causal interpretation. It is more likely that the possibility for such an interpretation was only partly achieved. Secondly, the current study was unable to determine the optimal level of cultural exposure for CHD risk, which types of cultural events that have the most beneficial effect or what is an ideal mix of cultural events to attend. These aspects represent research gaps that could be examined further in future studies. Finally, several variables, e.g., education, fitness and smoking, imply self-report bias. However, the limitations were balanced out by important strengths. The main strengths were the approach of using both MSM-Cox with IPW together with Cox proportional hazard regression models on longitudinal data including three measurements during a follow-up of 36 years in both men and women. Furthermore, the findings of an association between cultural exposure and CHD were in-line with previous findings on mortality from more traditional epidemiological design[5–8]. The consistency of our results with those from other studies further supports the validity and robustness of our findings. Finally, although the present study is not a randomised controlled trial, the use of MSM-Cox with IPW and the suggested dose-response effect may provide a more robust support for a potentially causal interpretation based on observation studies[11–15].

The high prevalence of asymptomatic atherosclerosis of the coronary arteries, a predecessor to CHD, is of major concern for health longevity[32] and emphasises the importance of preventive measures in broader societal measures.

In conclusion, this study demonstrates an association between attending cultural events and the incidence of CHD—the leading cause of mortality worldwide. This represents important information for preventive medicine focusing on CHD. Policymakers can use our findings to allocate health-promoting resources in society more efficiently. Although our findings cannot determine any causal effects, they are supportive of a potentially causal association between cultural exposure and CHD and emphasise the importance of consistent availability of cultural resources in human life. Future research on the specific protective mechanisms is warranted.

## Data availability

This study made use of national registers and, owing to legal concerns, data cannot be made openly available. Further information regarding the health registries is available from the following Swedish authorities: the Swedish National Board of Health and Welfare (https://www.socialstyrelsen.se/en/statistics-and-data/registers/ and *Statistics Sweden* (https://www.scb.se/en/ and https://www.scb.se/en/finding-statistics/statistics-by-subject-area/living-conditions/living-conditions/living-conditions-surveys-ulfsilc/).

## Code availability

The code used in the analysis has been made available online as Supplementary Data. The statistical software and standard methods used herein have been described in detail (with references) in the Methods. The statistical software R[20] was used for analyses in the MSM-Cox[11–15], STATA[22] was used for analyses in the Cox proportional hazard regression and the IPW R-package[23] was used to estimate IPW in the longitudinal data.

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

## Acknowledgements

F.J. was supported by non-commercial research grants, i.e., by governmental funding (Alf funding) for clinical research within the Swedish Public Healthcare, Region Skåne (Sweden). K.S. was also funded by grants from the Swedish Heart and Lung Foundation. The funding sources of the study were all non-commercial and had no role in the study design; the collection, analysis and interpretation of data; the writing of the report; or the decision to submit the paper for publication. We thank Patrick O'Reilly for his useful comments and language edits.

## Author contributions

All authors have approved the final version of the manuscript. Concept and visualisation: S.E.J., L.O.B. and K.S. Development of idea and design: all authors. Funding and resources: K.S. and F.J. Access and acquisition of data: K.S. Analysis and statistics: S.E.J. Tables: S.E.J. and F.J. Interpretation of data: all authors. Literature search and drafting of manuscript: F.J. and L.O.B. Critical revision of the manuscript for intellectual content: K.S. and S.E.J. The authors attest that all listed authors meet the authorship criteria and that no others meeting the criteria have been omitted.

## Funding

## Competing interests

The authors declare no competing interests.
