## [Peer Review File · Communications Medicine]

Reviewers' comments:

Reviewer #1 (Remarks to the Author):

The authors use inverse probability weighted (IPW) Cox regression models to account for potential time-varying confounding to explore the potential association between attending cultural events and coronary heart disease.

My major concern is that the authors seem to think that simply because they use somewhat advanced statistical methods they can ignore the fact that the study uses non-randomized data and is therefore potentially susceptible to confounding and other sources of bias.

This is most clear by their very districting need to constantly talk about causal effects of their associations – clearly there is an association between attending a cultural event and the risk of CHD. There is little reason to argue that this is causal however – which should not necessarily be a reason to reject the paper or lessen its message.

I appreciate this may not be clearly presented by the more methodological papers which present IPW and other MSM, but these methods make the strong assumption that ALL confounders have been measured (without error) and correctly modelled (e.g., that the association with the exposure is correctly modelled). As with all non-randomized studies there is no reason to expect this to be true, possibly it is approximately true but how would we know? By the authors own admission “confounding variables between cultural exposure and CHD has not been well-studied”. So clearly this assumption is unlikely to hold.

Aside from this exchangeability assumption, these models assume positivity (that there is an exposed and unexposed group for each level of the confounder), consistency (that all subjects respond the same for the same exposure value). The consistency assumption is likely an issue here. The authors use self-reported data on the frequency of attending cultural events, with cultural events diversely defined as attending an art museum, cinema, concert, sermons or theatres. I hope it goes without saying but why would one expect that attending a slipknot concert elicits the same CHD affect as going to a quite museum? The problem of consistency is further compounded by turning this data into a score and arbitrarily using quartiles to define low, middle or high level of cultural attendance – thus assuming that people within level (irrespective of the variability within each group, say 1 min of museum vs 1 hour) respond the same.

The authors perform a comparison against a more standard Cox model. From their explanation it is unclear if they used a time-varying Cox model. (see next point).

In the abstract the author say they used MSM to account for time-varying exposures. In fact standard Cox models can do this already – if used correctly. MSM such as IPW are used in settings when in addition to time-varying exposures, exposure may have an effect on confounders measured at a different time point. In this setting traditional “conditional” analysis would remove part of the treatment effect that is mediated through its effect on confounders, which may not be ideal.

In the abstract the authors make it sound like MSM is distinct from IPW, where of course IPW is a certain type of MSM.

Aside from the constant (and annoying) claim of causality, the paper is nicely done and potentially informative if one is willing to accept to more modest claim of association.

Reviewer #2 (Remarks to the Author):

The study by Johansson et al is very interesting study to understand the associations between attending cultural events in relation to future CVD risk. However, I have the following comments,

1. The way cultural exposure variable is defined is unclear and complicated. It would be helpful if authors can create a table and supplementary text that how different exposure estimates were computed to create the cultural exposure variable. Also, it is not clear how authors took into consideration when for example a participant at visit 1 has different inclination to exposure (e.g., a person visits cinema every week but if during visit 2 that person seldomly visits cinema) were taken into consideration.
2. Similarly, for the confounders -Marital status, physical exercise, educational level, how the change of these assessments during visits were taken into consideration.
3. How the regions were divided into large cities, middle large cities, and other regions? Did authors use some national guidelines to classify the regions into the levels that has been mentioned in the manuscript.
4. I wonder why age and sex interactions were introduced in the main analyses?
5. It would be helpful if authors can adjust analyses with the body mass index or general adiposity measures. or perform stratified analyses based on adiposity.
6. The causal relationship is a too strong statement ---. As the results are not based on a clinical trial research, so, I think causal relationship statement is very strong.
7. In Table 1, the way information is presented are unclear and confusing.
8. In table 2, it would be good to explain the results and also present p-values, number of participants in each group. How many incident CVD/CHD cases were? Why interaction of age × sex term was introduced. It would be great to present the results using marginal effect terms only. It is also helpful to present the results across both genders.

Minor comments

1. Not sure how these references (14 and 15 in the introduction section) are relevant

Author Responses

Malmö, 2023-02-26

Ref: COMMSMED-22-0185-T

Re: The causal association between attending cultural events and coronary heart disease – a longitudinal study with three measurements (1982–2017)

Referee expertise:

Referee #1: epidemiology, cardiovascular disease

Referee #2: epidemiology, cardiovascular disease

Reviewers' comments:

Reviewer #1 (Remarks to the Author):

1. The authors use inverse probability weighted (IPW) Cox regression models to account for potential time-varying confounding to explore the potential association between attending cultural events and coronary heart disease.

Response: *Thank you for your time and valuable feedback. We have responded to your comments below (such as the ones about the use of causal language and the need to clarify the methods section further).*

2. My major concern is that the authors seem to think that simply because they use somewhat advanced statistical methods they can ignore the fact that the study uses non-randomized data and is therefore potentially susceptible to confounding and other sources of bias.

This is most clear by their very districting need to constantly talk about causal effects of their associations – clearly there is an association between attending a cultural event and the risk of CHD. There is little reason to argue that this is causal however – which should not necessarily be a reason to reject the paper or lessen its message.

Response: *We agree that the causal language in this manuscript was too strong, and we have toned this down substantially. The causal language was used because Marginal Structural Modeling Cox (MSM-Cox) with inverse probability weights (IPW) is quite an established method to estimate (albeit not necessarily determine) causal effects in observational studies. However, we also agree that the study will still be susceptible for confounding and other sources of bias, which we have elaborated on further in the revised manuscript.*

Please see below for examples on how we have toned down the causal language:

Last paragraph in the introduction:

“For example, MSM-Cox has been used to estimate the causal average treatment effect from observational data similar to that of randomized controlled trials¹² in order to minimise, albeit not eliminate, the risk of residual confounding and certain forms of selection bias.^{11”}

Discussion:

First sentence:

“This longitudinal study attempted to assess a possible causal association between attending cultural events and coronary heart disease.”

Second paragraph, third sentence:

“This may be attributed to the use of weights in the MSM-Cox models with IPW that may further minimise, albeit not eliminate, the risk of residual confounding due to time-dependent covariates, which may act as both confounders and mediators.^{11-14,24”}

Limitations and strengths section:

“Therefore, we cannot determine whether all assumptions for MSM-Cox were fulfilled for a causal interpretation. It is more likely that the possibility for such an interpretation was only partly achieved.”

“Finally, although the present study is not a randomised controlled trial, the use of MSM-Cox with IPW and the suggested dose-response effect may provide a more robust support for a potentially causal interpretation based on observation studies.^{11-14,19”}

Last sentence and conclusion:

“Although our findings cannot determine any causal effects, they are supportive of a potentially causal association between cultural exposure and CHD and emphasise the importance of consistent availability of cultural resources in human life. Future research on the specific protective mechanisms is warranted.”

3. I appreciate this may not be clearly presented by the more methodological papers which present IPW and other MSM, but these methods make the strong assumption that ALL confounders have been measured (without error) and correctly modelled (e.g., that the association with the exposure is correctly modelled). As with all non-randomized studies there is no reason to expect this to be true, possibly it is approximately true but how would we know? By the authors own admission “confounding variables between cultural exposure and CHD has not been well-studied”. So clearly this assumption is unlikely to hold.

Response: *We agree with this comment and have now further emphasized that the use of MSM-Cox with IPW will not allow for firm causal inferences since it is not possible to fulfill all assumptions. Please see the description of the statistical methodology in the methods section and the first sentence in the last paragraph in the introduction section.*

4. Aside from this exchangeability assumption, these models assume positivity (that there is an exposed and unexposed group for each level of the confounder), consistency (that all subjects respond the same for the same exposure value). The consistency assumption is likely an issue here. The authors use self-reported data on the frequency of attending cultural events, with cultural events diversely defined as attending an art museum, cinema, concert, sermons or theatres. I hope it goes without saying but why would one expect that attending a slipknot concert elicits the same CHD affect as going to a quite museum? The problem of consistency is further compounded by turning this data into a score and arbitrarily using quartiles to define low, middle or high level of cultural attendance – thus assuming that people within level (irrespective of the variability within each group, say 1 min of museum vs 1 hour) respond the same.

Response: *We appreciate this comment and have revised the text in the manuscript in several places in order to reflect this. For example, in the paragraph describing the statistical methods, we do mention the assumptions in MSM-Cox with IPW but we also add that we judged that these assumptions for a causal association were only partly satisfied in our study.*

We have also revised the first sentence in the last paragraph in the introduction section to include that all assumptions must be satisfied for a causal inference:

“Marginal Structural Modelling Cox (MSM-Cox) with inverse probability weights (IPW) is an established method to estimate causal treatment effects in observational studies and provides a more robust assessment than conventional Cox models in the presence of time-varying variables, given that all assumptions are satisfied.^{11-14”}

We also agree that the consistency assumption could be an issue, especially in the use of an index since it cannot be assumed that all cultural events (used in the index) are consistent in their relationship with CHD. We have elaborated on this in the limitations in the Discussion:

“Secondly, the current study was unable to determine the optimal level of cultural exposure for CHD risk, which types of cultural events that have the most beneficial effect or what is an ideal mix of cultural events to attend. These aspects represent research gaps that could be examined further in future studies.”

For the rationale behind the use of quartiles, we have clarified that these categories were chosen because they were judged to be useful and illustrative in the interpretation of the results.

5. The authors perform a comparison against a more standard Cox model. From their explanation it is unclear if they used a time-varying Cox model. (see next point).

Response: *We apologize for being unclear. We used a time-varying Cox model in addition to the MSM-Cox with IPW (with time-varying weights). The manuscript has now been revised to include this information.*

6. In the abstract the author say they used MSM to account for time-varying exposures. In fact standard Cox models can do this already – if used correctly. MSM such as IPW are used in settings when in addition to time-varying exposures, exposure may have an effect on confounders measured at a different time point. In this setting traditional “conditional”

analysis would remove part of the treatment effect that is mediated through its effect on confounders, which may not be ideal.

Response: *We appreciate this comment and apologize for being unclear. We used a time-varying Cox model in addition to the MSM-Cox with IPW (with time-varying weights). The entire manuscript has now been revised to include this information.*

7. In the abstract the authors make it sound like MSM is distinct from IPW, where of course IPW is a certain type of MSM.

Response: *Thank you for pointing this out; it has been amended (please see above).*

8. Aside from the constant (and annoying) claim of causality, the paper is nicely done and potentially informative if one is willing to accept to more modest claim of association.

Response: *Thank you for this comment! The causal language has been toned down substantially. Please also see our response to comment 2 above. We appreciate your time and important comments on this manuscript.*

Reviewer #2 (Remarks to the Author):

The study by Johansson et al is very interesting study to understand the associations between attending cultural events in relation to future CVD risk. However, I have the following comments:

Response: *Thank you for your time and valuable feedback. We have responded to your comments below.*

1. The way cultural exposure variable is defined is unclear and complicated. It would be helpful if authors can create a table and supplementary text that how different exposure estimates were computed to create the cultural exposure variable. Also, it is not clear how authors took into consideration when for example a participant at visit 1 has different inclination to exposure (e.g., a person visits cinema every week but if during visit 2 that person seldomly visits cinema) were taken into consideration.

Response: *We apologize for being unclear. We have now expanded and clarified the description of how we constructed the cultural exposure variable. We also took into consideration the differences in exposure over time by using a time-varying Cox model in addition to the MSM-Cox with IPW (with time-varying weights). This means that we calculated a new score for the cultural attendance index at each of the three measurements during the follow-up period which allowed us to take into consideration the potential change in the cultural activities over time. We are, however, unsure about how we should create a table to elucidate this in a clear manner and prefer to only expand and clarify the text in the manuscript. For the revised description of the cultural exposure variable; please see the paragraph with the subheading 'Ascertainment of cultural exposure' in the methods section.*

2. Similarly, for the confounders -Marital status, physical exercise, educational level, how the change of these assessments during visits were taken into consideration.

Response: *We used time-varying Cox proportional hazard regression models and time-varying weights in MSM Cox with IPW. Thus, both Cox models allowed all variables (measured every eight years) to vary over time with the exception for sex (which did not change during the follow-up period). We apologize for not describing this well in the original manuscript; information that describes this more clearly has now been included in the manuscript.*

3. How the regions were divided into large cities, middle large cities, and other regions? Did authors use some national guidelines to classify the regions into the levels that have been mentioned in the manuscript.

Response: *The variable for regions was a predefined register variable created by Statistics Sweden and collected from the Total Population Register; this information has now been added in the manuscript.*

4. I wonder why age and sex interactions were introduced in the main analyses?

Response: *Age and sex interactions were included because sex had a modifying effect on the strength of the association between CHD and age. We have added a reference to this in the methods section, where this interaction has been described in another setting (<https://www.ahajournals.org/doi/10.1161/01.CIR.99.9.1165>).*

5. It would be helpful if authors can adjust analyses with the body mass index or general adiposity measures, or perform stratified analyses based on adiposity.

Response: *Unfortunately, body mass index or general adiposity measures were not collected in the study sample that we used in the present study.*

6. The causal relationship is a too strong statement ---. As the results are not based on a clinical trial research, so, I think causal relationship statement is very strong.

Response: *We agree that the causal language in this manuscript was too strong, and we have toned this down substantially. The causal language was used because Marginal Structural Modeling Cox (MSM-Cox) with inverse probability weights (IPW) is quite an established method to estimate causal effects in observational studies. However, we also agree that the study will still be susceptible for confounding and other sources of bias, which we have elaborated on further in the revised manuscript.*

7. In Table 1, the way information is presented is unclear and confusing.

Response: *We have revised the descriptive data in Table 1 as well as the headings and hope that these revisions have clarified what we are presenting in this Table.*

8. In table 2, it would be good to explain the results and also present p-values, number of participants in each group. How many incident CVD/CHD cases were? Why interaction of age \times sex term was introduced. It would be great to present the results using marginal effect terms only. It is also helpful to present the results across both genders.

Response: *We have revised the subheadings in all tables in order to clarify what we present in each table. We have added p-values in Table 2, number of participants in each exposure group over time in Table 1 and number of incidents in Table 2. Please see our response to comment 4 above that explains why an interaction of age \times sex term was introduced. This is the only interaction that was significant, and the addition of its interaction term means that it is not necessary to present the results by sex (although we are willing to do so if this still is requested). We also prefer to present the results from both Cox models since the use of two methodological approaches increases the robustness of our findings and the probability that our results are valid.*

Minor comments

1. Not sure how these references (14 and 15 in the introduction section) are relevant.

Response: *We agree, the sentence with the two references has been removed from the introduction section.*

REVIEWERS' COMMENTS:

Reviewer #1 (Remarks to the Author):

Many compliment to the authors for being receptive to my critical review of their original manuscript. The current revision has been greatly improved. In line with my previous concern on whether this research can robustly estimate a causal effect, I would strongly suggests to adjust the title and perhaps simply say (removing the word "causal"):

The association between attending cultural events and coronary heart disease – a longitudinal study with three measurements (1982–2017).

Reviewer #2 (Remarks to the Author):

I have no further comments. I am satisfied with the reviewers input.

Author Responses

Malmö, 2023-04-05

Regarding: Final revisions for manuscript COMMSMED-22-0185A

Previous title: The causal association between attending cultural events and coronary heart disease – a longitudinal study with three measurements (1982–2017)

Final title: A longitudinal study of the association between attending cultural events and coronary heart disease

REVIEWERS' COMMENTS:**Reviewer #1 (Remarks to the Author):**

Many compliment to the authors for being receptive to my critical review of their original manuscript. The current revision has been greatly improved. In line with my previous concern on whether this research can robustly estimate a causal effect, I would strongly suggests to adjust the title and perhaps simply say (removing the word "causal"):

The association between attending cultural events and coronary heart disease – a longitudinal study with three measurements (1982–2017).

Response: *Thank you for your time and valuable feedback when reviewing our manuscript; we have removed the word causal from the title. The title now reads (at editorial request):*

A longitudinal study of the association between attending cultural events and coronary heart disease

Reviewer #2 (Remarks to the Author):

I have no further comments. I am satisfied with the reviewers input.

Response: *Thank you for your time and valuable feedback when reviewing our manuscript.*